# Bottlenecked Transformers: Periodic KV Cache Consolidation for Generalised Reasoning

**Adnan Oomerjee**
University College London
Huawei Noah's Ark, UK
adnan.oomerjee.22@ucl.ac.uk

**Zafeirios Fountas**
Huawei Noah's Ark, UK
zafeirios.fountas@huawei.com

**Haitham Bou-Ammar**
Huawei Noah's Ark, UK
University College London
haitham.ammar@huawei.com

**Jun Wang**
University College London
jun.wang@ucl.ac.uk

## Abstract

Transformer LLMs have been shown to exhibit strong reasoning ability that scales with inference-time compute, most prominently through token-space "thinking" chains of thought. A growing line of work pushes extra computation into the model's latent space, which we term Auxiliary Latent-Space Computation (ALSC). Existing ALSC methods largely fall into three buckets: (i) token-mediated latent rollouts, (ii) residual/activation steering, and (iii) memory (KV) compression. An underexplored alternative is memory consolidation/reconsolidation, two processes in the brain that are responsible for stabilising newly formed memory traces, and, upon recall, transiently rendering established traces plastic such they can integrate new contextual information before restabilising. In Transformer LLMs, this can be seen as analogous to performing in-place rewrites of new KV segments, and rewrites of recalled past segments. In this work, we give a theoretical justification as to why memory (re)consolidation via KV cache rewrites is beneficial for improved reasoning. We do this through the lens of Information Bottleneck (IB) theory, which posits that model generalisation emerges from an optimal balance between input information compression and retention of predictive information in latent representations. We then introduce the Bottlenecked Transformer, which augments a backbone LLM with a Cache Processor, an auxiliary Transformer that performs periodic, non-causal, in-place KV rewrites at newline-delimited reasoning step boundaries. The Processor consolidates recently written KV entries and reconsolidates a small, top-$k$ attention-selected set of prior entries. We evaluate our Bottlenecked Transformer architecture on math reasoning benchmarks. Our model sees consistent performance gains over vanilla Transformers and pause-token augmented baselines, with gains of up to +6.6pp for selected tasks/backbones.

## 1 Introduction

Transformer-based large language models (LLMs) have achieved strong results in retrieval, pattern recognition, and knowledge extraction (Brown et al., 2020; Chowdhery et al., 2022). With carefully engineered prompts/post-training, they can also display nontrivial reasoning behaviours (Wei et al., 2023; Shao et al., 2024; DeepSeek-AI et al., 2025). A critical development that has significantly advanced Transformer LLMs is the discovery that reasoning performance scales strongly with inference-time compute. The most widely applied example of this has been seen in "reasoning" models, which generate verbal chains of thought before giving a final answer (Wei et al., 2023).

A growing body of work extends this idea to algorithms that allow LLMs to perform additional compute during generation directly in a latent space rather than the token space. We refer to these as *Auxiliary Latent-Space Computation* (ALSC) methods, which facilitate computation over internal continuous states during inference without emitting intermediate natural-language tokens, doing so in addition to (or in place of) the standard one-forward-pass-per-token decoding strategy. In

this work we focus on sequence-level ALSC: operators that intervene between decoding steps to transform the model's KV cache and/or final hidden state before the LM head, which constitute the model's internal representation of a processed sequence. Auxiliary Latent-Space Computation is potentially more efficient than strict autoregressive decoding as latent embeddings can encode semantics more compactly than token sequences. Additionally, such processes align more closely with human cognition, in which thought does not occur as an endless verbal monologue, but contains nonverbal stretches of conceptual processing that proceeds without recruiting the language system (Alderson-Day & Fernyhough, 2015; Fedorenko & Varley, 2016; Monti et al., 2012).

Prior sequence-level ALSC approaches primarily fall into three categories: token-space latent stepping, activation-space edits, and cache-operators (most commonly compressive schemes such as pruning, merging, or summarising KV entries). An underexplored direction is incorporating processes for memory *consolidation* and *reconsolidation* in the neuroscientific sense. Consolidation is a process in the brain where new memory traces are stabilised upon formation. Reconsolidation refers to rewrites of recalled memories: when a stored memory is reactivated, it can briefly enter a plastic state in which it can be modified before restabilising, allowing it to be updated and recontextualised with new salient information (Lee, 2009; Hupbach et al., 2007).

In this paper, we explore consolidation and reconsolidation in Transformer LLMs from both a theoretical and architectural standpoint. We adopt a working interpretation in which the KV cache serves as the model's memory and (re)consolidation is realised through periodic in place edits to that memory during generation. We first offer an information-theoretic justification for why periodically reprocessing the model's working memory (KV cache) should aid generalisation from the lens of Information Bottleneck theory; concretely, we show that in autoregressively trained models, the KV cache is incentivised to preserve information from the sequence history that is unnecessary for future sequence-level prediction, potentially hindering generalisation. We then introduce the *Bottlenecked Transformer*, which augments a pretrained backbone with a *Cache Processor*, a small Transformer that periodically rewrites recent memories (consolidation) and selectively recalled KV entries (reconsolidation) in-place, without dimensional compression. Our architecture is shown in Figure 1. Note that whilst our aim is to implement a mechanism functionally analogous to consolidation and reconsolidation in the brain, the underlying biological processes are richer and more complex than our computational abstraction. Empirically, we yield consistent gains over vanilla Transformers across seven mathematical reasoning benchmarks and multiple backbones.

## 2 PRELIMINARIES

In all following sections, we denote random variables by uppercase letters (e.g. $X, Y, Z$) and realisations by the lowercase letters (e.g. $x, y, z$).

### 2.1 STATE-SPACE FORMULATION OF DECODING

Autoregressive decoding in a Transformer can be viewed as a state-space process, where the model's key–value (KV) cache is its memory state. Formally, let $x_t$ be the input token at decoding step $t$. For a model with $L$ layers, we let $h_t$ denote the KV cache (covering tokens $0 : t$), $o_t \in \mathbb{R}^d$ the final-layer residual stream. We then model the next-token decoding process as:

$$h_t \equiv \big\{ (k_{0:t}^{(\ell)}, v_{0:t}^{(\ell)}) \big\}_{\ell=1}^{L}$$

$$(h_t, o_t) = f_{\text{LLM}}(h_{t-1}, x_t), \qquad p(x_{t+1} \mid h_t, o_t) = \text{softmax}\big(f_{\text{head}}(o_t)\big).$$

This view treats $h_t$ as the sequence-level latent state that mediates future predictions, while $o_t$ summarizes the current step's computation. Under a vanilla LLM, updating $h_t$ amounts to per-layer appends of the key–value vectors produced for the incoming token $x_t$.

### 2.2 SEQUENCE-LEVEL AUXILIARY LATENT-SPACE COMPUTATION

We call an *auxiliary latent-space computation* (ALSC) method any inference-time procedure that performs extra computation over internal continuous states adjacent to the standard forward pass of a backbone LLM. We focus on *sequence-level* ALSC that acts on $(h_t, o_t)$ via

$$(h', o') = \mathcal{T}(h_t, o_t),$$

invoked according to a schedule $s(t) \subseteq \mathbb{N}$ (e.g., periodic every $m$ steps or event-triggered). After applying $\mathcal{T}$, decoding resumes from the transformed state:

$$p(x_{t+1} \mid h', o') = \mathrm{softmax}\big(f_{\mathrm{head}}(o')\big).$$

## 3 RELATED WORK

We classify sequence-level ALSC works into three execution pathways: (i) Token-mediated, (ii) residual-operator, and (iii) cache-operator.

**(i) Token-mediated.** These methods instantiate $\mathcal{T}$ to be an LLM (often the backbone LLM itself), and operate via an internal micro-sequence of latent tokens, thereby lengthening the cache and updating $(h_t, o_t)$ via a standard forward pass. Basic variants inject pause or filler tokens during decoding (Goyal et al., 2024; Pfau et al., 2024). Cache Deliberation uses an external coprocessor (often initialized from the backbone) to produce latent embeddings conditioned on $h_t$ and append them to the cache (Liu et al., 2024a). Other approaches recycle the model's last hidden state as a continuous latent fed back as the next input embedding, taking multiple latent steps without emitting text until termination (Hao et al., 2024; Shen et al., 2025; Su et al., 2025).

**(ii) Residual-operator.** A complementary line defines $\mathcal{T}$ to only modify the current hidden representation $o_t$ before the LM head, leaving $h_t$ unchanged. *Activation steering* adds structured directions to $o_t$ (or selected layers) to influence style, stance, or safety (Turner et al., 2024), including Contrastive Activation Addition (Panickssery et al., 2024) and exemplar-derived "style vectors" (Konen et al., 2024). Recent work targets sparse features for precision and interpretability, e.g., SAE-targeted steering (Chalnev et al., 2024), operations directly in SAE latent space (FGAA) (Soo et al., 2025), and broader SAE-based frameworks (He et al., 2025).

**(iii) Cache-operator.** In these methods, $\mathcal{T}$ operates solely to transform the memory $h_t$. Here, the cache is transformed directly between decoding steps to control what information remains accessible. Existing methods are predominantly centred on memory compression for long context tasks. *Eviction/pruning* methods retain high-utility entries using importance or heavy-hitter policies, preserving pivotal tokens and stable "sink" anchors (Zhang et al., 2023; Xiao et al., 2024). *Merging/aggregation* mechanisms fuse entries into representatives under a memory budget (Zhang et al., 2024; Wang et al., 2024). *Recurrent architectures* summarize older activations into compact memories or memory banks (Transformer-XL, Compressive Transformers, RMT) (Dai et al., 2019; Rae et al., 2019; Bulatov et al., 2022). *Selective recall* architectures externalise long histories to memory banks and re-inject salient slices on demand (Fountas et al., 2024).

**Positioning.** Memory (re)consolidation as we interpret it belongs to the cache-operator family, but differs from the predominantly compression-oriented approaches above. In-place memory rewrites under (re)consolidation do not necessarily entail reduction in memory footprint. Rather, our goal is to demonstrate how KV rewrites may improve reasoning performance in Transformer LLMs.

## 4 MOTIVATION AND THEORY

In this section, we give a theoretical analysis as to why a mechanism for (re)consolidation via KV rewrites is likely to improve on performance on reasoning tasks in Transformer decoder-only LLMs, from the perspective of Information Bottleneck Theory. All proofs are given in Appendix A.

### 4.1 THE INFORMATION BOTTLENECK METHOD

The Information Bottleneck (IB) is a framework for optimising some latent variable Z to be maximally informative of some output variable Y and minimally informative of an input variable X, via the objective:

$$\mathcal{L}[p(z|x)] = I(X; Z) - \beta \, I(Z; Y) \tag{1}$$

subject to $Y \leftrightarrow X \leftrightarrow Z$. Here $\beta > 0$ balances *information compression* via lowering $I(X; Z)$ with *relevance* $I(Z; Y)$ (Tishby et al., 2000). Controlling $I(X; Z)$ has been proven to bound test set generalization error as $\epsilon \leq O\big(\sqrt{(I(X; Z) + 1)/n}\big)$ for i.i.d. data (Kawaguchi et al., 2023), with strong empirical results that suggests this extends to non-i.i.d. time series data (Feng et al., 2024; Liu et al., 2024b; Ullmann et al., 2023; Choi & Lee, 2024).

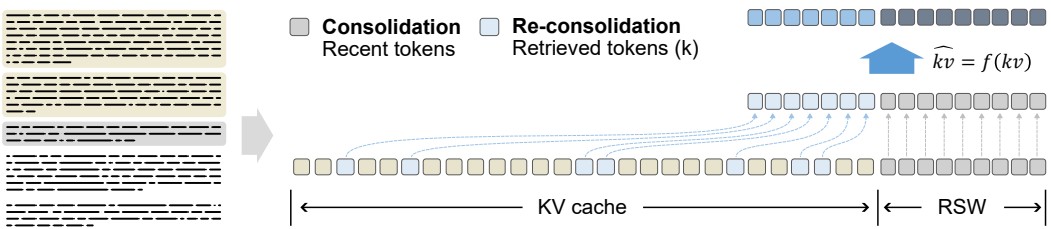

Figure 1: Bottlenecked Transformer architecture, consisting of a backbone LLM processing/generating tokens, and Transformer Cache Processor that rewrites KV entries. The Cache Processor is invoked each time a newline token is generated (marking the end of a reasoning step). When invoked, recent tokens (from the recent step window in grey) and $k$ retrieved tokens beyond the RSW (in blue) are passed in parallel to the Cache Processor, and rewritten in-place.

## 4.2 INFORMATION BOTTLENECKS AND DEEP LEARNING

In practice, one can approximate true distributions $p(z|x)$ and $p(y|x)$ by parameterized distributions $p_\phi(z \mid x)$ and $p_\psi(y \mid z)$. The joint model $p_\theta(x, y, z)$ now factorizes as:

$$p_\theta(x, y, z) = p(x)p_\phi(z|x)p_\psi(y|z) \tag{2}$$

and we optimize $\mathcal{L}$ with respect to $\theta = (\phi, \psi)$. To ground these ideas, we now introduce three formal definitions that capture the essential properties and ordering of information bottlenecks in neural networks.

**Definition 4.1** (Neural Information Bottleneck). *Let $\mathcal{M}_\theta$ be a neural network parameterised by $\theta$, with input/output variables $(X, Y)$, and let $Z$ be a latent variable within the model. Then, $Z$ is an information bottleneck in $\mathcal{M}_\theta$ if and only if $Z$ satisfies the Markov chain $X \to Z \to Y$.*

**Definition 4.2** (Ordering of Bottlenecks in Neural Networks). *Let $\{Z_i\}_{i \in I}$ be a set of distinct information bottlenecks in $\mathcal{M}_\theta$. We say that a bottleneck $Z_j$ is* deeper *than another bottleneck $Z_i$, denoted by $Z_i \prec Z_j$, if and only if $X \to Z_i \to Z_j \to Y$.*

**Definition 4.3** (Terminal Bottleneck). *Let $\mathcal{Z}^{\mathcal{M}_\theta}$ be the set of all information bottlenecks in $\mathcal{M}_\theta$. Then $\hat{Z} = \max \mathcal{Z}^{\mathcal{M}_\theta}$ is denoted the terminal bottleneck in $Z^{\mathcal{M}_\theta}$.*

The notion of ordering of bottlenecks allows for the observation that the complexity $I(X; Z)$ of any arbitrary bottleneck $Z$ is bounded by $I(X; \hat{Z})$, the complexity of the terminal bottleneck, formalised in Lemma 4.1.

**Lemma 4.1.** *Let $\mathcal{M}_\theta$ be a model parameterized by $\theta$, with input/output variables $(X, Y)$, and let $\mathcal{Z}^{\mathcal{M}_\theta}$ be the set of information bottlenecks in $\mathcal{M}_\theta$, with $\hat{Z}$ defining the terminal bottleneck in $\mathcal{M}_\theta$. Then $I(X; Z) \geq I(X; \hat{Z})$ for any bottleneck $Z \in \mathcal{Z}^{\mathcal{M}_\theta}$.*

**Implicit Information Compression During SGD** Even without an explicit IB loss, noise inherent to stochastic gradient descent (SGD) has been shown to implicitly minimise $I(X; Z)$ in neural networks. During training, after an initial "fitting" phase, SGD hase been shown to enter a low-signal-to-noise "diffusion" regime in which gradient noise dominates, systematically compressing input information in hidden representations (Shwartz-Ziv & Tishby, 2017; Butakov et al., 2024).

## 4.3 IB OBJECTIVE FOR GENERALISED LANGUAGE REASONERS

The problem of learning a generalised language-based reasoner can be formulated as one of learning a generalised sequence to sequence model via the IB objective. Given an input $X = S_{0:n}$ (a reasoning history of $n$ transitions), and $Y = S_{n+1}$ (a subsequent reasoning step), we seek to train a model $\mathcal{M}_\theta$ aiming to predict $S_{n+1}$ given $S_{0:n}$. Under the IB method, given a neural information bottleneck $Z$ in $\mathcal{M}_\theta$ that sequentially mediates $(S_{0:n}, S_{n+1})$, we define the IB objective:

$$\theta^* = \arg\min_\theta I(S_{0:n}; Z) - \beta I(Z; S_{n+1}) \tag{3}$$

Under this objective, a realised latent $z$ should act as *abstraction* of the reasoning history to infer a generalised state of a partial solution, from which some logical rule of inference can be applied.

## 4.4 Analysis of Information Bottlenecks in Decoder-Only Transformer LLMs

Our first main result is given in Theorems 4.1 and 4.2. We show that in a decoder-only Transformer, given a input sequence $S_{0:n}$ and output sequence $S_{n+1}$, the KV cache and last hidden state computed from $S_{0:n}$ forms the terminal bottleneck $\hat{Z}$ mediating these sequences, and autoregressive training maximises both $I(S_{0:n}; \hat{Z})$ and $I(\hat{Z}; S_{n+1})$.

**Theorem 4.1** (KV-Cache and Final Hidden State as Seq-to-Seq Terminal Bottleneck). *Let $\mathcal{M}_\theta^{LLM}$ be a decoder-only Transformer language model parameterized by $\theta$, with input/output sequence variables $(S_{0:n}, S_{n+1})$. We define the information bottleneck $C_{0:n}$ as:*

$$C_{0:n} = (K_{0:n}, V_{0:n}, O_n)$$

*where $K_{0:n}$ and $V_{0:n}$ are represent keys and values computed from $S_{0:n}$ across all heads/layers, and $O_n$ is the final hidden-state vector of the last token of $S_{0:n}$ prior to the model's final logit projection. Then $C_{0:n} = \hat{Z}$, the terminal bottleneck in $\mathcal{M}_\theta^{LLM}$.*

**Theorem 4.2** (Autoregressive Training Encourages high $I(S_{0:n}; \hat{Z})$ and $I(\hat{Z}; S_{n+1})$). *Let $s_{0:N}$ be a complete reasoning trace drawn from $p(s_{0:N})$. For some $n$ (where $0 \leq n < N$), we define $(s_{0:n}, s_{n+1})$ to be an input/output pair corresponding to an incomplete reasoning history and ground-truth next reasoning step. Let $\mathcal{M}_\theta^{LLM}$ be a decoder-only Transformer that maps input $s_{0:n}$ to KV cache/final hidden state $c_{0:n} = (k_{0:n}, v_{0:n}, o_n)$ via a determinstic mapping $f_\phi$, where $\phi \subset \theta$:*

$$c_{0:n} = f_\phi(s_{0:n}) \tag{4}$$

*Let $L(\theta)$ be the expected next step log-likelihood to maximise (computed via negative cross entropy) with respect to parameters $\theta$:*

$$L(\theta) = \mathbb{E}_{p(s_{0:N})}\left[\sum_{n=0}^{N-1} \log p_\theta(s_{n+1} \mid s_{0:n})\right] \tag{5}$$

*Then we can show two bounds on $L(\theta)$:*

$$L(\theta) \leq \sum_{n=1}^{N} I(S_{0:n}; C_{0:n}) - \sum_{n=0}^{N-1} H(S_{n+1}|S_{0:n}) \tag{6}$$

$$L(\theta) \leq \sum_{n=0}^{N-1} I(C_{0:n}; S_{n+1}) - H(S_{n+1}) \tag{7}$$

Under Theorem 4.2, since $L(\theta)$ acts as a bound on each term (where the entropy terms are fixed under a particular dataset), maximisation of $L(\theta)$ thus acts to encourage raising both mutual information components $I(C_{0:n}; S_{n+1})$ and $I(S_{0:n}; C_{0:n})$. When we consider the tokenwise view, with $C_{0;t}$ representing tokens at timesteps 0 to $t$, we can see that $C_{0:t}$ contains, as sub-states, each earlier cache $C_{0:i}$ for $i < t$, and so contains sufficient information to recover the full collection of next-token predictors $\{p_\theta(S_{i+1} \mid C_{0:i})\}_{i<t}$ realised along the sequence. Consequently, the final cache encodes a high-fidelity, step-by-step predictive trace of the right-shifted tokens $(S_1, \ldots, S_t)$, rather than a single compressed summary of the past.

Combined with Lemma 4.1 and Theorem 4.1, this implies that autoregressive training encourages internal sequence representations that are *minimally* compressive of their inputs as well as maximally predictive of future outputs.

## 4.5 Comparisons with RNNs and Cache Compression Methods

Transformers excel at retrieval-style tasks, due to their effectively unbounded memory, while RNNs and structured state-space models often outperform on problems requiring systematic rule application or OOD generalisation (Deletang et al., 2023; Liu et al., 2023; Wen et al., 2025). Due to the hard sequence-level bottleneck imposed by RNNs via their fixed-size hidden state, latent representations are forced to be reprocessed and compressed at every time step, whereas standard Transformers' ever-growing cache removes this constraint completely. This leads to mutual information between given inputs ($X$) and these compressed latents ($Z$) that is reduced relatively to the one between latents $Z$ and predicted outputs ($Y$), compared to Transformers. See Fig. 2.B for a conceptual illustration of this comparison.

Existing cache-operator methods have largely been explored from the perspective of compression via memory footprint reduction (see Section 3). As illustrated conceptually in Fig. 2 A, these algorithms tend to reduce not only the information retained about the input ($I(X; Z)$), but also indiscriminately reduce predictive information ($I(Z; Y)$), thereby moving towards a region of lower generalised performance on benchmark tasks. Crucially, these methods lack a reprocessing step designed to selectively compress $I(X; Z)$ while preserving or enhancing $I(Z; Y)$. Consequently, without a mechanism to substantially improve predictive efficiency ($\frac{I(Z;Y)}{I(X;Z)}$), as depicted in Fig. 2.B, these techniques offer limited improvements in generalisability.

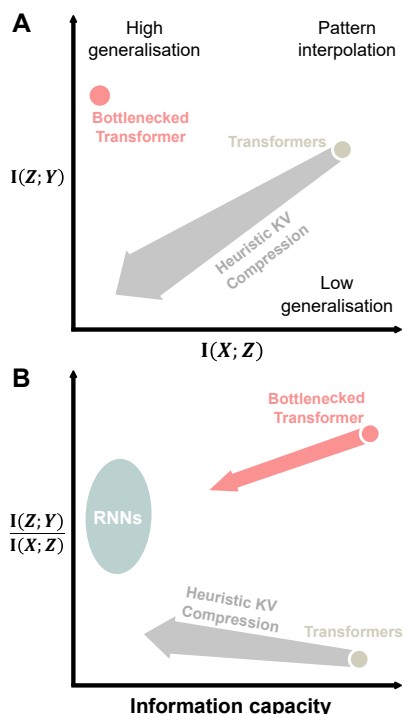

Figure 2: Conceptual illustration. (**A**) Bottlenecked Transformers balance input compression $I(X; Z)$ with predictive information $I(Z; Y)$ for high generalisation. (**B**) This achieves superior predictive efficiency $I(Z; Y)/I(X; Z)$ vs. capacity over other methods.

### 4.6 ARCHITECTURAL SOLUTIONS

Our analysis shows that the KV cache (together with the final hidden state) forms the terminal bottleneck $\hat{Z}$ and, in practice, carries extraneous detail from processed sequences. This motivates an inference-time mechanism that rewrites KV entries in place, producing a new bottleneck $\hat{Z}' = \mathcal{T}(\hat{Z})$ with an increase in predictive efficiency $I(\hat{Z}'; Y)/I(X; \hat{Z}')$. By the data-processing inequality, any such transformation results in $I(X; \hat{Z}) \geq (X; \hat{Z}')$. By training $T$ to minimise future prediction error, we preserve or improve $I(\hat{Z}'; Y)$. Conceptually, we interpret this as analogous to consolidation/reconsolidation: selectively reprocessing working memory to discard irrelevant information and maintain salient information. We focus on rewriting KVs rather than the final hidden state as the cache is the component principally responsible for retaining the extraneous sequence information. We apply no dimensionality reduction in rewritten KVs so as to avoid indiscriminate reduction in predictive information that plagues compression methods.

## 5 BOTTLENECKED TRANSFORMERS

**Overview.** Here we introduce the *Bottlenecked Transformer*. We augment a pretrained decoder-only Transformer $\mathcal{M}_\theta^{\mathrm{LLM}}$ with an external Cache Processor $\mathcal{T}_\omega^{\mathrm{proc}}$, a neural module (smaller than the backbone) that periodically rewrites KV cache entries in-place during autoregressive generation. A illustration of our architecture can be seen in Figure 1.

**Processor Invocation and Mechanism.** During generation, immediately after some reasoning step $s_n$ completes (detected by emission of a newline token), the Processor is invoked to rewrite cache entries. Decoding then resumes conditioned on the rewritten cache. We design our rewrite mechanism to be analogous to memory consolidation/reconsolidation in the brain, and implement a selective mechanism for rewriting cache entries. When invoked, the Processor rewrites (i) cache entries corresponding to the most recent segment $s_n$, and (ii) the top $k$ entries from the prior step history $s_{0:n-1}$ by attention mass with the recent segment $s_n$. These components realise mechanisms that are respectively analogous to consolidation and reconsolidation: new memories within a recent step window (RSW) of variable length $R$ undergo a stabilisation process, and recalled memories are rewritten in light of new information. Formally, we designate the set of recalled and recent KV entries as $(k_{(s)}, v_{(s)})$. All other KV entries are left unchanged. A more detailed formulation of our selection mechanism can be found in Appendix B.

**Cache Processor Architecture.** For a backbone $\mathcal{M}_\theta^{\mathrm{LLM}}$ with $L$ layers and $H$ heads, the Cache Processor consists of $L$ small Transformer blocks $\{\mathcal{T}_\omega^{\mathrm{proc},(\ell)}\}_{\ell=1}^L$, one aligned to each backbone layer. Block $\ell$ operates only on the corresponding layer's selected KV entries $(k_{(s)}^{(\ell)}, v_{(s)}^{(\ell)})$. We first convert the selected key–value pairs into "KV-tokens" by concatenating across all heads for that layer, and

| Base LLM | Method | GSM8K | MATH | SVAMP | Task TheoremQA | LogiQA | Gaokao-MathQA | GSM-Hard |
|---|---|---|---|---|---|---|---|---|
| Llama 3.2 1B | SFT | 29.80 | 11.76 | 38 | 8.84 | 15.36 | 3.70 | 7.13 |
| | SFT w/ pause tokens | 30.02 | 11.34 | 41.6 | 7.22 | 13.36 | 1.71 | 7.96 |
| | SFT w/ latent rollout | 24.41 | 9.66 | 32.8 | 8.97 | 15.36 | 1.13 | 5.31 |
| | Bottlenecked Transformer (ours) | **32.97** | **12.72** | **44.6** | **10.84** | **19.05** | 3.99 | 7.96 |
| Llama 3.1 8B | SFT | 70.28 | 31.88 | 77.7 | 15.26 | 20.74 | 4.84 | 19.41 |
| | SFT w/ pause tokens | 69.22 | 31.52 | 77.2 | **15.93** | 20.12 | **5.98** | 18.88 |
| | SFT w/ latent rollout | 4.02 | 2.80 | 7.80 | 5.89 | 0.00 | 0.00 | 0.61 |
| | Bottlenecked Transformer (ours) | **71.87** | **31.96** | **78.4** | **15.93** | **23.81** | 3.99 | **19.93** |
| Qwen 3 0.6B | SFT | 53.75 | 26.68 | 60.7 | 14.32 | 23.04 | **5.70** | 19.56 |
| | SFT w/ pause tokens | 52.92 | 26.76 | 60.3 | **14.73** | 21.50 | 5.13 | 19.26 |
| | SFT w/ latent rollout | 47.23 | 20.28 | 57.70 | 12.32 | 24.27 | 3.42 | 17.66 |
| | Bottlenecked Transformer (ours) | **57.01** | **29.08** | **65.4** | **14.73** | **26.57** | 5.41 | **20.55** |
| Llama 3.2 3B | SFT | 46.78 | 18.40 | 55.5 | 10.71 | **22.12** | 3.13 | 11.45 |
| | SFT w/ pause tokens | 48.07 | 18.00 | 55.9 | 12.05 | 17.67 | **4.84** | 11.6 |
| | SFT w/ latent rollout | 42.46 | 15.28 | 52.0 | 11.65 | 12.90 | 2.56 | 10.61 |
| | Bottlenecked Transformer (ours) | **51.33** | **20.90** | **59.4** | **14.73** | 20.12 | 3.99 | **12.28** |

Table 1: Accuracy (%) on seven mathematical reasoning benchmarks across four backbones and three configurations: SFT, SFT+pause (16 pause tokens after the prefix), SFT+latent rollout (16 rollout tokens after the prefix), and Bottlenecked Transformer (ours; frozen SFT backbone augmented with Cache Processor). Scores are pass@1 under greedy decoding. Bold indicates the best result within each backbone.

project them via a learnable matrix into the Processor's hidden state space:

$$x^{(\ell)} = \left(k_{(s)}^{(\ell)}, v_{(s)}^{(\ell)}\right) \in \mathbb{R}^{(k+R) \times 2Hd_k}, \tag{8}$$

$$u^{(\ell)} = x^{(\ell)} W_{\text{in}}^{(\ell)}, \qquad W_{\text{in}}^{(\ell)} \in \mathbb{R}^{2Hd_k \times d_p}. \tag{9}$$

The sequence $u^{(\ell)}$ (consisting of recalled and recently cached memories) is then processed in parallel by a small Transformer block without causal masking, such that selected KV entries may be updated with globally available information. The block's output is projected back via learnable matrix to the KV dimensionality. Finally, we apply a gated, in-place residual rewrite to the selected KV entries.

$$\tilde{\Delta}^{(\ell)} = \mathcal{T}_\omega^{\text{proc},(\ell)}\left(u^{(\ell)}\right) \tag{10}$$

$$\left(\Delta_k^{(\ell)}, \Delta_v^{(\ell)}\right) = \tilde{\Delta}^{(\ell)} W_{\text{out}}^{(\ell)} \qquad W_{\text{out}}^{(\ell)} \in \mathbb{R}^{d_p \times 2Hd_k} \tag{11}$$

$$k_{(s)}^{(\ell)} \leftarrow k_{(s)}^{(\ell)} + \sigma\left(g^{(\ell)}\right) \Delta_k^{(\ell)} \qquad v_{(s)}^{(\ell)} \leftarrow v_{(s)}^{(\ell)} + \sigma\left(g^{(\ell)}\right) \Delta_v^{(\ell)} \tag{12}$$

Here $g^{(\ell)} \in \mathbb{R}$ is a learnable, layer-wise scalar gate initialized small and $\sigma$ denotes the logistic function. The gate mitigates early drift in model capabilities, i.e., large, destabilizing cache changes before the Processor has learned useful updates.

**Training.** Learning proceeds in two stages. In the first stage, the backbone $\mathcal{M}_\theta^{\text{LLM}}$ undergoes SFT on reasoning trajectories with the standard next-token cross-entropy objective. In the second stage, the backbone is frozen and only the processor parameters $\omega$ are updated. Each training sequence $s_{0:N}$ is split into individual reasoning steps $(s_0, \ldots, s_N)$. For step $s_n$, the backbone first processes the tokens in that step to append new KVs to the cache. The Processor is then invoked, selecting $(k_{(s)}^{(\ell)}, (k_{(s)}^{(\ell)})$ at each layer and applying the in-place rewrite. Cross entropy loss for the next reasoning step $s_{n+1}$ is then computed, conditioned on the rewritten cache, and backpropagated through the Processor. We truncate BPTT across step boundaries, such that the Processor is trained solely to rewrite the cache in a way that improves prediction of the next reasoning step.

Note that we do not implement an IB-style loss function; our goal is to realise a plausible mechanism for (re)consolidation in Transformer LLMs, supported by our theoretical findings that periodic memory rewrites may improve generalisation. The rewritten cache realises a new sequence-level terminal bottleneck $\tilde{Z}$. Training the Processor to minimise the cross entropy loss of the entire next reasoning step is equivalent to maximising $I(S_{n+1}; \tilde{Z})$. In other words, $\tilde{Z}$ is trained solely to improve prediction of future sequences, with no requirement for the rewritten entries to retain unnecessary information for reconstructing their input sequence. Moreover, whilst we do not explicitly include a compression term for minimisation of $I(S_{0:n}, \tilde{Z})$, removal of pressure to maximise this quantity (as we showed to occur in vanilla Transformers) opens a pathway for implicit minimisation of this term via noise injection from SGD (as described in Section 4.2).

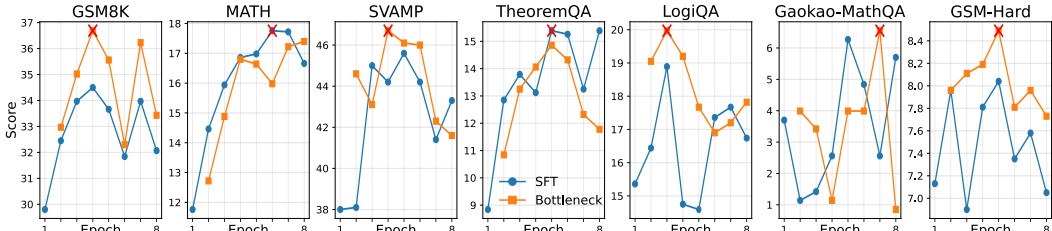

Figure 3: Epoch-matched comparison of *SFT@N* and *Bottleneck@N* across seven tasks. The backbone is SFT-trained for 8 epochs with per-epoch checkpoints; Bottleneck@$N$ uses checkpoint $N-1$ plus one Processor epoch, and curves plot accuracy versus total epochs $N$. The red $\times$ marks the highest score for each task across both model variants and all $N$.

## 6 EXPERIMENTS

### 6.1 PERFORMANCE ON MATHEMATICAL REASONING TASKS

We evaluate the Bottlenecked Transformer on a set of mathematical reasoning tasks, choosing this domain as it offers an easily verifiable testbed for for observing improved reasoning generalisation. We compare four settings: (i) a vanilla LLM fine-tuned for one epoch on a math dataset (SFT), (ii) a pause-token baseline trained identically but includes 16 pause tokens appended after each question prefix (SFT+pause, following prior convention), (iii) a latent rollout model (inspired by Coconut (Hao et al., 2024)) which performs an n-step latent rollout directly in the token space by feeding the final hidden state back into the model without decoding to tokens, and (iv) our Bottlenecked Transformer, which freezes the one-epoch SFT model as a backbone and trains the Cache Processor for one epoch using the procedure in Section 5. We use SFT as a standard Transformer baseline and SFT with pause and SFT with latent rollout as token-mediated ALSC baselines. We omit other ALSC variants (residual/cache operators) as these primarily target style/behavior control or memory footprint reduction rather than generalisation. All models are trained on 128k examples from OpenMathInstruct-2, a large synthetic mix of GSM8K/MATH-style questions (Toshniwal et al., 2024). We evaluate on seven benchmarks: GSM8K, MATH, SVAMP, TheoremQA, LogiQA, Gaokao-Math, and GSM-Hard. Six are mathematical reasoning tasks; LogiQA is a logical reasoning task included to test transfer beyond mathematics. For all experiments, we fix Processor hidden size $d_p = 512$, intermediate size $2240$, 16 heads per Processor block, selective reconsolidation uses $k = 32$. Detailed hyperparameters can be found in Appendix D.1.

Results are given in Table 1. Across backbones and tasks, the Bottlenecked Transformer improves over both baselines in almost all cases. Gains are strongest on in-distribution math benchmarks (GSM8K, MATH, SVAMP, GSM-Hard): e.g., Llama-3.2 1B on SVAMP (+6.6 points, 38.0→44.6), Llama-3.2 3B on GSM8K (+4.6, 46.78→51.33), Qwen-3 0.6B on MATH (+2.4, 26.68→29.08), and Llama-3.1 8B on LogiQA (+3.1, 20.74→23.81). On the more out-of-distribution QA-style tasks, improvements generally persist (e.g., TheoremQA matches or exceeds baselines on all backbones), with one notable exception: LogiQA on Llama-3.2 3B where plain SFT is slightly higher (22.12 vs. 20.12). The main underperformance is Gaokao-MathQA, where baselines often win (e.g., Qwen-0.6B and Llama-3.1 8B), consistent with a distribution/language shift (Chinese) beyond the Cache Processor's training exposure. By contrast, the pause-token baseline shows variable and often lower performance than plain SFT when used only at fine-tuning (e.g., consistent drops on Llama-3.1 8B and Qwen-3 0.6B), with only occasional wins such as TheoremQA at 8B or Gaokao-Math on some backbones. This mirrors findings from the original pause token paper, which showed reliable gains only when paired with continued pretraining before SFT. Additionally, the latent rollout baseline typically underperforms even the pause token baseline, which is consistent with results seen in the original Coconut paper, wherein the model performed slightly worse than a Vanilla model but saw improved efficiency (fewer tokens needed per answer). Performance degradation is especially bad for the Llama 3.1 8B model, which sees severe model destabilisation under continuous latent rollouts.

### 6.2 EPOCH-MATCHED TRAINING BUDGET ABLATION

To compare extra SFT with cache (re)consolidation under the same training budget, we align models by the total number of training epochs seen. We first train a backbone with SFT for 8 epochs, saving a checkpoint after each epoch. For every checkpoint, we freeze the backbone and train a

| $k$ | GSM8K | MATH | SVAMP | TheoremQA | LogiQA | Gaokao-Math | GSM-Hard |
|-----|-------|------|-------|-----------|--------|-------------|----------|
| 16 | 32.07 | 13.12 | 43.20 | 10.04 | 19.05 | 3.70 | 7.58 |
| 32 | 32.97 | 12.72 | 44.60 | 10.84 | 19.05 | 3.99 | 7.96 |
| 64 | 33.43 | 12.94 | 44.20 | 10.04 | 19.20 | 2.28 | 7.96 |
| 128 | 33.05 | 13.20 | 43.30 | 9.64 | 19.05 | 3.13 | 7.73 |
| 256 | 33.05 | 13.34 | 43.30 | 9.50 | 17.81 | 2.28 | 7.58 |

Table 2: Top-$k$ ablation of the bottleneck model across tasks (backbone: Llama 3.2 1B). Each column is color-scaled from red (lowest) through yellow (middle) to green (highest), with softened tones.

| $R$ | GSM8K | MATH | SVAMP | TheoremQA | LogiQA | Gaokao-Math | GSM-Hard |
|-----|-------|------|-------|-----------|--------|-------------|----------|
| 16 | 31.69 | 12.92 | 42.70 | 9.77 | 18.89 | 4.27 | 7.88 |
| 32 | 32.45 | 12.23 | 43.20 | 10.98 | 17.97 | 2.85 | 7.81 |
| 48 | 31.99 | 12.84 | 42.30 | 11.38 | 18.13 | 3.13 | 7.96 |
| 64 | 32.07 | 12.18 | 43.40 | 12.05 | 17.97 | 3.13 | 7.58 |
| 96 | 32.22 | 13.02 | 44.00 | 10.58 | 19.35 | 5.13 | 7.96 |

Table 3: Ablation over recent-step window size $R$ for the Bottlenecked Transformer (backbone: Llama 3.2 1B). The Cache Processor is invoked once at the end of the prompt and then every $R$ tokens, so $R$ controls the length of the local segment consolidated at each update. Performance is broadly stable across $R$, with slight gains for moderate windows.

Cache Processor for one additional epoch. We then compare SFT@$N$ (pure SFT for $N$ epochs) against Bottleneck@$N$ built from checkpoint $N-1$ plus one Processor epoch (both variants have seen $N$ epochs). We use a Llama 3.2 1B backbone, with same Cache Processor configuration as in Section 6.1. Across all seven tasks (Fig. 3), Bottleneck@$N$ outperforms SFT@$N$ on most $N$ for GSM8K, GSM-Hard, SVAMP, and LogiQA, and the best score attained on these tasks over any $N$ is achieved by a Bottleneck model. Two consistent exceptions are MATH and, to a lesser extent, TheoremQA, where SFT@$N$ tends to be higher; additionally, Gaokao-MathQA mostly favors SFT@$N$ at a given $N$, although the single best score over all $N$ is still achieved by a Bottleneck model. A plausible reason is that these settings require sustained access to precise symbolic/theorem or language-specific details, and step-boundary top-$k$ reconsolidation ($k{=}32$) may down-weight earlier formula tokens or non-English cues that remain predictive.

### 6.3 RECONSOLIDATION BUDGET ($k$) ABLATION

We ablate the Processor's attention-guided selection budget by varying the number of prior positions $k$ that are reconsolidated per layer at each Processor invocation, holding all other settings fixed (backbone: Llama 3.2 1B; identical training/evaluation protocol as Section 6.1). For each $k$ we train a separate Processor and report accuracy on the seven benchmarks. Table 6.2 summarizes results. Across all tasks except MATH, moderate budgets ($k \approx 32$ to $k \approx 64$) are generally optimal. In contrast, MATH benefits from larger budgets, with best scores at $k \approx 128$ or 256. This likely reflects that MATH contains harder problems with longer solutions and stronger long-range dependencies. It also offers a plausible explanation for the Bottleneck model's weaker MATH performance in the training budget experiment (Section 6.2), where the reconsolidation window was fixed at $k = 32$.

### 6.4 RECENT STEP WINDOW ($R$) ABLATION

We also ablate the size of the recent-step window $R$ by invoking the Cache Processor once at the end of the prompt and then at every fixed $R$ tokens during generation, so that $R$ directly controls how many of the most recent tokens are consolidated at each call (Table 6.2). Across benchmarks, performance remains relatively stable over a broad range of $R$, with mild gains for moderate to larger windows (e.g., $R \approx 64$–96) and small drops when consolidation is restricted to very short windows. This suggests that the Processor benefits from access to a reasonably sized local context, but does not require fine-grained, per-token updates to yield gains. Together with the top-$k$ reconsolidation ablation, these results indicate that our memory (re)consolidation mechanism is robust to the precise update schedule, so long as it can periodically reshape a medium-horizon segment of the working memory rather than attempting to track every token verbatim.

### 6.5 PROCESSOR REWRITE MAGNITUDES

We measure how strongly the Cache Processor modifies the KV cache by tracking cosine distances between entries before and after each rewrite. On GSM8K with the Llama 3.2 1B Bottlenecked

Transformer, we compute mean cosine distance at every processor invocation for (i) the top-$k$ recalled tokens, (ii) the recent-step window (RSW), and (iii) all rewritten entries. Results are shown in Figure 4. Across all three groups, value vectors undergo nontrivial updates, while key vectors remain almost unchanged, indicating that the Processor mainly edits the contents of memory rather than its addressing. Rewrite magnitudes are largest at early processing steps and then settle into a stable plateau after roughly ten invocations, showing that the Processor does not collapse to the identity map but applies consistent moderate adjustments throughout generation. Layer–head heatmaps of value-vector distances show that edits are concentrated in the earliest layers, with only small changes in middle and later layers. This suggests that the Processor learns to reshape low-level representations that then propagate forward through the backbone, rather than rewriting deep layers directly.

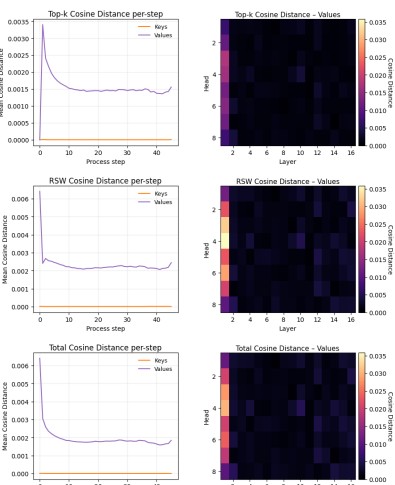

Estimating $I(X; Z)$ for a high-dimensional, variable-length KV cache is intractable in our setting, so we use rewrite magnitudes as a qualitative proxy for how the Processor reshapes the working memory state. Our observation of systematic shifts in value vectors away from their teacher-forced encodings indicates that the model is restructuring the content of selected memories. Because the Processor is trained solely through next-step prediction loss, these local, persistent edits suggest that past information is being reorganised while maintaining what is useful for future tokens. From the Information Bottleneck perspective, such prediction-preserving, non-identity updates are naturally associated with reducing redundant input detail and making more efficient use of the bottleneck; here we treat rewrite magnitudes as an indirect, qualitative signal of this process rather than a direct estimate of information-theoretic quantities.

Figure 4: Cache Processor rewrite magnitudes on GSM8K. Left: per-invocation mean distances for top-$k$, recent-step window, and all rewritten tokens. Right: layer–head heatmaps of mean cosine distance between pre- and post-Processor value vectors.

# 7 DISCUSSION AND FUTURE WORK

Our work has explored the gap in cache-operator ALSC systems that pertains to our interpretation of memory (re)consolidation, giving both a theoretical justification as to why this beneficial in decoder-only Transformer LLMs and empirical verification via an architecture that improves mathematical reasoning performance. Here we discuss limitations of our method.

Training the Processor solely through next-step cross-entropy can produce high-variance, poorly localized credit assignment, providing weak supervision for cache rewrites, making it challenging for the model to escape its strong local optimum. Training a model from scratch may alleviate this issue. Additionally, we do not include an explicit information–theoretic objective for compression via reduction of $I(X; Z)$: any information compression can only arise from the data processing inequality or SGD noise. Whilst direct MI estimation in a high-dimension cache is challenging, a promising route is controlled noise injection into selected KV entries followed by iterative denoising/refinement, which constitutes a mapping that reduces $I(X; Z)$ while preserving predictive structure $I(Z; Y)$ (by the data–processing inequality and denoising-as-regularization). Such a mechanism would essentially constitute iterative latent reasoning in the model's memory space; past works exploring this idea in non-LLM-based frameworks have yielded promising results (Du et al., 2024).

Regarding our interpretation/implementation of consolidation and reconsolidation, neuroscientific literature indicates that these are related but partially distinct processes: consolidation unfolds over hours to days with systems-level reorganization and sleep-driven replay, whereas reconsolidation is a brief, retrieval-induced window in which a reactivated trace becomes labile and then re-stabilises (Dudai et al., 2015; Stickgold & Walker, 2007). In light of this, our single, online Processor collapses two modes that in biology differ in triggers and timescales; a closer analogue would pair an offline, replay-style consolidator with an online, retrieval-contingent reconsolidator. Additionally, reconsolidation appears to depend on prediction error at retrieval, i.e., a mismatch is often required to open the plastic window, suggesting that surprise/PE gating (rather than a fixed newline trigger) would be more suitable for determining when reconsolidation should occur (Exton-McGuinness et al., 2015; Fernández et al., 2016). More closely aligning future (re)consolidation architectures with these biological mechanisms may yield substantial gains over our current models.

## ETHICS STATEMENT

We adhere to the ICLR Code of Ethics. Our study uses publicly available, licensed datasets and synthetic math corpora; no human subjects or private data were involved.

## REPRODUCIBILITY STATEMENT

We detail all training and evaluation settings (datasets, preprocessing, hyperparameters, model sizes, and decoding) in Appendix D, and provide proofs for theoretical claims in Appendix A.

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

## APPENDIX

## A PROOFS

### A.1 PROOF OF LEMMA 4.1

*Proof.* Under the non-trivial case where $Z \neq \hat{Z}$, by Definition 4.2, we have that:

$$X \to Z \to \hat{Z} \to Y$$

Under the Data Processing Inequality, we thus have:

$$I(X; \hat{Z}) \leq I(X; Z).$$

$\square$

### A.2 PROOF OF THEOREM 4.1

*Proof.* Assume for contradiction that there exists some bottleneck $Z'$ strictly deeper than $C_{0:n}$ (i.e., $C_{0:n} \prec Z'$). By definition of the partial order, we could discard $C_{0:n}$ when predicting $S_{n+1}$, so

$$p_\theta(S_{n+1} \mid C_{0:n}, Z') = p_\theta(S_{n+1} \mid Z').$$

However, under a decoder-only Transformer, under arbitrary input/output sequence variables $(S_{0:n}, S_{n+1})$, decoding of $S_{n+1}$ must be conditioned on $C_{0:n}$, as $C_{0:n}$ is a projection of the input sequence ($S_{0:n}$ and is trained to contain all information necessary to predict $S_{n+1}$. Furthermore, under this architecture, no further processing occurs on $C_{0:n}$ once it has been constructed. Hence, $C_{0:n}$ in its entirety cannot be discarded and replaced by some variable Z', contradicting $Z' \prec C_{0:n}$. Therefore, $C_{0:n}$ is the maximal element in the set of bottlenecks, and thus it is the terminal bottleneck. $\square$

## A.3 PROOF OF THEOREM 4.2

*Proof.* Since $c_{0:n} = f_\phi(s_{0:n})$ is a deterministic mapping under a vanilla decoder-only Transformer, we can write our objective function as:

$$L(\theta) = \mathbb{E}_{p(s_{0:N}, c_{0:N})}\Big[\sum_{n=0}^{N-1} \log p_\theta\big(s_{n+1} \mid s_{0:n}\big)\Big] \tag{13}$$

$$= \mathbb{E}_{p(s_{0:N}, c_{0:N})}\Big[\sum_{n=0}^{N-1} \log p_\theta\big(s_{n+1} \mid c_{0:n}\big)\Big] \tag{14}$$

$$= \sum_{n=0}^{N-1} \mathbb{E}_{p(s_{n+1}, c_{0:n})}\Big[\log p_\theta\big(s_{n+1} \mid c_{0:n}\big)\Big] \tag{15}$$

For two random variables $A$, $B$ with samples $(a, b)$ from joint distribution $p(A, B)$ and approximate distribution $q(A, B)$, we see that:

$$\mathbb{E}_{p(a,b)}\left[\log q(a|b)]\right] \equiv \mathbb{E}_{p(a,b)}\left[\log p(a|b) + \log \frac{q(a|b)}{p(a|b)}\right] \tag{16}$$

$$\equiv -H(A|B) - \mathbb{E}_{p(b)}\left[D_{KL}\left[p(a|b) \,\|\, q(a|b)\right]\right] \tag{17}$$

$$\equiv I(A; B) - H(A) - \mathbb{E}_{p(b)}[D_{KL}\left[p(a|b) \,\|\, q(a|b)\right]] \tag{18}$$

$$\leq I(A; B) - H(A) \tag{19}$$

Which implies that:

$$L(\theta) \leq \sum_{n=0}^{N-1} I(C_{0:n}; S_{n+1}) - H(S_{n+1}) \tag{20}$$

We can also write $L(\theta)$ as:

$$L(\theta) = \mathbb{E}_{p(s_{0:N}, k_{0:N}, v_{0:N})}\Big[\sum_{n=0}^{N-1} \log p_\theta\big(s_{n+1} \mid s_{0:n}, k_{0:n+1}, v_{0:n+1}\big)\Big] \tag{21}$$

$$= \sum_{n=0}^{N-1} \mathbb{E}_{p(s_{0:n}, k_{0:n+1}, v_{0:n+1})}\Big[\log p_\theta\big(s_{n+1} \mid s_{0:n}, k_{0:n+1}, v_{0:n+1}\big)\Big] \tag{22}$$

$$= -\sum_{n=0}^{N-1} H(S_{n+1}|S_{0:n}, K_{0:n+1}, V_{0:n+1}) \tag{23}$$

$$= \sum_{n=0}^{N-1} I(S_{n+1}; K_{0:n+1}, V_{0:n+1}|S_{0:n}) - H(S_{n+1}|S_{0:n}) \tag{24}$$

$$\leq \sum_{n=0}^{N-1} I(S_{n+1}; C_{0:n+1}|S_{0:n}) - H(S_{n+1}|S_{0:n}) \tag{25}$$

$$\leq \sum_{n=0}^{N-1} I(S_{0:n+1}; C_{0:n+1}) - H(S_{n+1}|S_{0:n}) \tag{26}$$

$$= \sum_{n=1}^{N} I(S_{0:n}; C_{0:n}) - \sum_{n=0}^{N-1} H(S_{n+1}|S_{0:n}) \tag{27}$$

Thus, combining Equations 20 and 27 we have:

$$2L(\theta) \leq \sum_{n=1}^{N} I(S_{0:n}; C_{0:n}) + \sum_{n=0}^{N-1}\Big[I(C_{0:n}; S_{n+1}) - H(S_{n+1}) - H(S_{n+1}|S_{0:n})\Big]$$

$$L(\theta) \leq \frac{1}{2}\Big[\sum_{n=1}^{N} I(S_{0:n}; C_{0:n}) + \sum_{n=0}^{N-1}\Big[I(C_{0:n}; S_{n+1}) - H(S_{n+1}) - H(S_{n+1}|S_{0:n})\Big]\Big]$$

$$\square$$

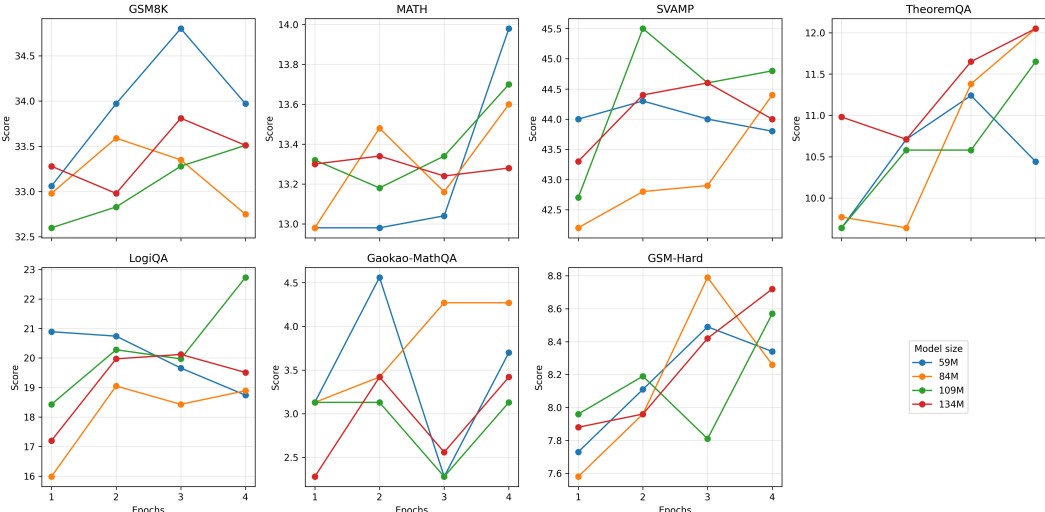

Figure 5: Size ablation of the Cache Processor on a frozen Llama 3.2 1B backbone, showing per-epoch performance of each variant on each task.

## B  CONSOLIDATION/RECONSOLIDATION MECHANISM

Let $J_n = \mathrm{Idx}(s_n)$ denote the token indices of a just-completed step and $P_{<n} = \mathrm{Idx}(s_{<n})$ the indices of all prior tokens. For each layer $\ell \in \{1, \dots, L\}$ we construct an index set

$$\mathcal{I}_n^{(\ell)} \;=\; J_n \;\cup\; \mathrm{TopK}_n^{(\ell)}(P_{<n})$$

where $\mathrm{TopK}_n^{(\ell)}(P_{<n})$ contains the $k$ prior positions with the largest attention mass with the current step. Writing $A^{(\ell,h)} \in \mathbb{R}^{t \times t}$ for the backbone's attention matrix at layer $\ell$, head $h$, over the prefix $s_{\leq n}$, the mass assigned by the tokens of $s_n$ to a prior index $i \in P_{<n}$ is

$$\alpha_i^{(\ell)} \;=\; \frac{1}{|J_n|\, H} \sum_{h=1}^{H} \sum_{j \in J_n} A_{j,i}^{(\ell,h)}, \qquad \mathrm{TopK}_n^{(\ell)}(P_{<n}) \;=\; \arg\operatorname*{topk}_{i \in P_{<n}} \alpha_i^{(\ell)},$$

where $H$ denotes the number of attention heads. Only the entries at $\mathcal{I}_n^{(\ell)}$ are modified by the Processor; all other cache positions remain unchanged. This realises (i) consolidation of recently written context, and (ii) reconsolidation of the few most recalled KVs during the recently written step.

## C  CACHE PROCESSOR SIZE ABLATION

In this experiment, we ablate the Cache Processor size and training duration using a Llama 3.2 1B backbone, by varying the Processor feedforward intermediate width, holding depth, number of heads, selection policy (RSW + top-$k$), and all training hyperparameters fixed (same as in Section 6.1, see details in Section D. The backbone is trained for one epoch of SFT on OpenMathInstruct-2. We train four Processors on this backbone (of 59M, 84M, 109M, 134M parameters). Each Processor is trained for four epochs, and we evaluate at the end of every epoch.

Results are shown in Figure 5. Increasing training duration generally improves performance of models on all tasks, with this effect being most apparent on the MATH, TheoremQA and GSM-Hard tasks, which typically contain the hardest problems. For other tasks, performance beyond one epoch of training is more variable across all models, often plateauing early. This suggests that additional training may not be beneficial in cases where downstream tasks are simpler. Additionally, there is no clear winner in model size across all tasks. This suggests that training so as to make full use of the model capacity is challenging. We hypothesise that this is due to poor credit assignment from next-step supervision, making it challenging to escape the strong local optimum that the backbone resides in, as discussed in Section 7.

# D  REPRODUCIBILITY

## D.1  MAIN RESULTS

Here we list details for reproducibility of our main experimental run in Section 6.1.

**Training details**    . We train on the first 128k examples from the 1M variant of the OpenMathInstruct-2 dataset (Toshniwal et al., 2024). For all runs, we use a batch size of 128 with learning rate of $1e - 4$. We use a constant LR with no warmup for all runs except for experiments with Llama 3.1 8B, where we use a warmup ratio of 0.05 and cosine LR scheduling. We truncate training sequences to a max length of 512 tokens.

**Evaluation details.**    For all evaluation runs, we employ greedy decoding, truncating responses to a maximum length of 2048. We evaluate on seven tasks:

- **GSM8K**: Grade-school math word problems requiring multi-step arithmetic reasoning and short numeric answers. (Cobbe et al., 2021)

- **MATH**: Competition-style mathematics problems (e.g., algebra, geometry, number theory, counting) with formal, multi-step solutions. Hendrycks et al. (2021)

- **SVAMP**: Simple arithmetic word problems rewritten with semantic variations to test robustness to superficial cues. (Patel et al., 2021)

- **TheoremQA**: Question answering that requires recalling, understanding, or applying mathematical theorems and their conditions. (Chen et al., 2023)

- **LogiQA**: Multiple-choice logical reasoning and argument analysis questions modeled after civil service exam items. (Liu et al., 2020)

- **Gaokao-MathQA**: Math question answering drawn from China's Gaokao (college entrance) exams, often involving symbolic manipulation and problem solving. (Zhong et al., 2024)

- **GSM-Hard**: A harder subset of grade-school math problems with much larger numbers, designed to stress multi-step reasoning beyond standard GSM8K difficulty.(Gao et al., 2022)

**Pause token baseline configuration.**    During training, we instantiate 16 pause tokens in the embedding table. During training/generation, we append these 16 tokens to the end of the question prefix. We perform SFT via standard cross entropy loss on response completions.

**Bottlenecked Transformer Configuration.**    For all backbones, we fix the Processor design across backbones (one block per backbone layer; hidden size $d_p$=512; MLP intermediate size 2240; 16 heads per block, fixing reconsolidation budget $k = 32$. Table D.1 shows parameter counts for the Processor for each backbone, these vary because the projection layers to/from KV-space and number of Processor blocks depend on the backbone's hidden dimension, number of attention heads, and number of layers.

| Backbone | Processor Params |
|---|---|
| Llama 3.2 1B | 88.69M |
| Llama 3.2 3B | 184.63M |
| Llama 3.1 8B | 211.01M |
| Qwen 3 0.6B | 184.63M |

Table 4: Cache-Processor parameter counts per backbone for mathematical reasoning performance experiment.

## D.2  ABLATIONS

For these experiments, where applicable, we use the same training/architectural configuration for SFT/Bottleneck models as is detailed in Section D.1, using a Llama 3.2 1B Backbone.

### D.3 COMPUTATIONAL OVERHEAD

For a Bottlenecked Transformer consisting of a Llama 3.2 1B backbone with an 89M parameter Cache Processor, setting $k = 32$, memory footprint during the Cache Processor training stage is approximately 6x that of performing full parameter SFT, owing to the chunked training process which prevents parallelism for entire training examples, induces extra padding tokens, as well as the high computational cost of processing a large number of entries in a high dimensional KV cache. As such, wall clock time for a one-epoch training run for the Cache Processor is approximately 20x longer than performing SFT on the backbone. Note that this is partially an engineering issue: our method is currently incompatible with efficient attention methods such as Flash Attention.

During evaluation, memory footprint of the Bottlenecked Transformer with above configuration is approximately 25% higher than a vanilla Transformer, with a 45% increase in wall clock time, for an eval batch size of 16 in both cases. This relative reduction compared to training is due to the fact that during generation, the Cache Processor is invoked infrequently (once every reasoning step).

## STATEMENT ON THE USE OF LLMS

During manuscript preparation, we used large language models for editing, phrasing suggestions, and to assist in literature search. No analyses, results, proofs, or figures were produced by LLMs; all technical content, experiments, and conclusions are our own.

