# OpenReview forum: "Bottlenecked Transformers: Periodic KV Cache Consolidation for Generalised Reasoning"
_ICLR.cc/2026/Conference — ICLR 2026 Poster_

### Official Review · Reviewer_kt4M · 2025-10-15

**Soundness:** 4
**Presentation:** 4
**Contribution:** 4
**Rating:** 8
**Confidence:** 3

**Summary:**

This paper introduces the Bottlenecked Transformer, an architecture designed to improve the reasoning capabilities of decoder-only LLMs. The authors first provide a theoretical motivation using Information Bottleneck (IB) theory, arguing that standard autoregressive training incentivizes the model's KV cache to retain unnecessary information from the input sequence, which can hinder generalization. To address this, they propose augmenting a frozen backbone LLM with a lightweight "Cache Processor." This processor is a separate, non-causal Transformer that is invoked periodically (at newline characters) to perform in-place rewrites of the backbone's KV cache. The rewrite mechanism is inspired by the neuroscience concepts of memory consolidation (acting on recent KV entries) and reconsolidation (acting on a small, attention-selected set of prior entries). The processor is trained to rewrite the cache in a way that improves the prediction of the next reasoning step. The authors demonstrate that this approach yields consistent performance gains on seven mathematical reasoning benchmarks across four different backbone models.

**Strengths:**

**Strong Theoretical Foundation:** The primary strength of this work is its grounding in Information Bottleneck (IB) theory. The authors provide a formal proof that vanilla Transformers are constrained in their ability to form optimal sequence representations for generalization.

**Novel and Bio-Inspired Architecture:** The concept of a separate Cache Processor that performs periodic, in-place KV cache rewrites is highly novel. The design, explicitly inspired by the neural mechanisms of memory consolidation and reconsolidation, is both elegant and intuitive.

**Comprehensive Empirical Validation:** The authors validate their method across seven different mathematical and logical reasoning benchmarks and, crucially, with four different open-source backbone models of varying sizes. The consistent performance improvements over both vanilla SFT and a pause-token baseline demonstrate the robustness and general applicability of the approach.

**Insightful Ablation Studies:** The paper includes well-designed ablation studies. The epoch-matched comparison effectively shows that an epoch of Processor training is a better use of compute than an additional epoch of SFT. The ablation on the reconsolidation budget (k) provides valuable insights into how task characteristics interact with the model's memory mechanism

**Weaknesses:**

**Significant Computational Overhead:** The practical utility of the method is currently hampered by its high computational cost. The authors report that training the Cache Processor is ~20x slower than standard SFT, and inference is ~45% slower on a 1B parameter model. While acknowledged as a potential engineering issue, this overhead is a major barrier to adoption.

**Weak and Indirect Supervision:** The Cache Processor is trained only via the cross-entropy loss of the next reasoning step. As the authors note, this provides a high-variance and poorly localized signal for learning optimal cache rewrites, which may explain why performance doesn't always improve with Processor size or training duration.

**Simplistic Invocation Trigger:** The Processor is invoked every time a newline token is generated. While simple to implement, this heuristic is not adaptive. It may trigger too often in contexts where newlines are frequent but semantically unimportant, or not often enough in dense reasoning chains that lack newlines.

**Narrow Domain of Evaluation:** The experiments are exclusively focused on mathematical reasoning. While this is an excellent testbed for logical generalization, it's unclear how the core principle—compressing the past to better predict the future—would transfer to other domains like creative writing or summarization, where preserving rich, verbatim details from the context is often essential.

**Questions:**

1. Regarding the computational overhead: Could you elaborate on the specific engineering challenges that prevent compatibility with methods like Flash Attention? Is there a clear path forward to mitigating the significant training and inference latency, or is it a fundamental cost of the non-causal, in-place rewrite operation?

2. The supervision signal for the Processor seems to be a key challenge. Have you considered or experimented with auxiliary loss functions to provide a more direct learning signal? For example, could a variational IB objective be approximated, or could a contrastive loss be used to encourage the rewritten cache state to be more predictive than the original state?

3. The choice to trigger on newline tokens is a simple heuristic. How sensitive are the results to this specific choice? For instance, what would happen if the trigger was changed to a period or another punctuation mark, and have you considered more adaptive triggers based on model uncertainty or prediction error, as suggested in your discussion?

4. The paper makes a compelling case for memory consolidation in the context of mathematical reasoning. How do you see this mechanism applying to domains where reasoning is less about logical abstraction and more about maintaining narrative consistency or tracking complex entity relationships, such as in long-form story generation? Could the "bottlenecking" process be detrimental in such cases?

---

> ### Author Response · Authors · 2025-11-24
> **Response to reviewer kt4M**
>
> We thank the reviewer for the thoughtful and detailed feedback, and for the positive assessment of our paper. Below we respond to the questions, which also address the main weaknesses raised in the review.
>
> Regarding **Question 1 (computational overhead)**, we agree that the current training and inference cost is a major practical limitation. Our goal in this paper was to explore whether cache rewrites are beneficial at all, even with a relatively unoptimised first implementation. One point we will clarify in the paper is that FlashAttention *is* used within the Cache Processor itself, but not in the backbone during Processor training. The training schedule requires chunk-wise passes over reasoning steps with intermittent padding, which leads to situations where the query length differs from the KV length; this is not compatible with our FlashAttention implementation in the backbone stack. In contrast, within the Processor the KV and query lengths always match at each attention call, so FlashAttention works there. We view the resulting overhead as largely an engineering artefact of this training setup rather than a fundamental cost of non-causal, in-place rewrites, and we plan to explore alternative scheduling and kernel designs in future variants to reduce both training and inference latency.
>
> For **Question 2 (supervision signal and auxiliary losses)**, we agree that relying solely on next-step cross-entropy with truncated BPTT gives a weak and indirect learning signal for the Processor, and we now highlight this as a significant limitation. Direct MI estimation on high-dimensional Transformer caches over long sequences is technically challenging, and in preliminary experiments simple MI-style regularisers and naive variational IB objectives (without strong priors on the cache structure) did not improve performance and sometimes harmed it. Our next variants therefore focus on more explicit target cache states: for example, using gradient-based optimisation on the cache for individual examples to find states that minimise loss, and then training the Processor to map to those targets with a more direct objective such as MSE. This would provide a more local and principled supervision signal for the rewrites. While we have not yet integrated such objectives into the current paper’s results, we agree this is a crucial direction and will clarify both the limitations of our present supervision scheme and our plans for stronger objectives.
>
> Concerning **Question 3 (newline triggering vs. adaptive triggers)**, our current choice is deliberately simple: in the math datasets we use, solutions are formatted so that each line corresponds to a distinct intermediate reasoning step, so newline tokens align closely with semantic “step boundaries”. This makes newline a reasonable surrogate trigger in our setting, and in practice we found it stable. We have not yet systematically evaluated alternative token-based triggers such as periods or other punctuation, nor uncertainty- or prediction-error-based triggers, and we will clarify this in the paper. As discussed in the main text and by the other reviewers, we expect that more adaptive triggers (based on surprise, model confidence, or learned boundary detectors) could further improve performance and robustness, and we see this as a natural extension rather than something we claim to have solved here.
>
> Finally, for **Question 4 (application beyond mathematical reasoning)**, we appreciate the reviewer’s point that compressing the past to better predict the future may interact differently with tasks like long-form story generation, where preserving rich episodic detail and narrative consistency is often crucial. Our experiments are restricted to mathematical reasoning, and we now state this scope explicitly. From the IB perspective, overly aggressive reduction of $I(X;Z)$ could indeed be detrimental in domains where verbatim recall of earlier context is important, and it is plausible that different tasks have different “optimal compression” regimes. We anticipate that in such settings, one would need to tune or adapt the consolidation policy (e.g., window size, reconsolidation budget, gating) so that the Processor abstracts where appropriate but preserves detailed traces for entities and narrative threads. Exploring this trade-off in creative writing, summarisation, and long-form dialogue is an exciting direction for future work, and we are careful not to claim universal benefits of bottlenecking in the current paper.
>
> We again thank the reviewer for their insightful comments and suggestions, which we believe help clarify both the current scope and the most promising avenues for improving the architecture.

---

### Official Review · Reviewer_cuPq · 2025-10-18

**Soundness:** 4
**Presentation:** 3
**Contribution:** 3
**Rating:** 6
**Confidence:** 3

**Summary:**

The paper proposes Bottlenecked Transformers, augmenting a decoder-only LLM with a small Cache Processor that periodically rewrites selected KV-cache entries at newline-delimited “reasoning step” boundaries to mimic consolidation/reconsolidation and improve generalized reasoning. The authors motivate this with an Information Bottleneck (IB) analysis arguing that vanilla Transformers’ KV caches act as terminal, minimally compressive bottlenecks and thus preserve extraneous input information that can hinder generalization.

**Strengths:**

* **Clear theoretical motivation**: The IB framing (Theorems 4.1–4.2) formalizes the KV cache + final hidden state as a terminal bottleneck and links autoregressive training to maximizing both $I(X;Z)$ and $I(Z;Y)$, motivating selective rewrites.

* **Simple, modular mechanism**: A lightweight, layer-aligned processor rewrites (i) recent-step KVs and (ii) top-k recalled past entries by attention mass; gating stabilizes updates. The schedule is practical (trigger on newline).

* **Consistent empirical gains**. Across backbones/tasks, the model usually beats SFT and pause-token baselines; Table 1 and Fig 3 show broad improvements and useful diagnostics.

**Weaknesses:**

* **Scope & novelty relative to cache operators**: While the reconsolidation framing is fresh, the core operation (transform selected cache entries) is close to existing cache-edit/compression lines; novelty hinges on scheduling/selection rather than a fundamentally new cache objective.

* **Supervision signal may be weak**: The processor is trained only via next-step cross-entropy with truncated BPTT, which the authors note causes credit-assignment issues; no explicit IB/MI control is used.

* **Mixed results on OOD and symbol-heavy tasks**: MATH/LogiQA/Gaokao occasionally favor SFT; performance is sensitive to k and language/domain shift, suggesting limited generality without careful tuning.

**Questions:**

* **Ablations on triggers**: Newline ≠ reasoning step in many datasets. How do results change with alternative triggers (e.g., surprise/prediction-error gates, punctuation, tokens-per-step), or with learnable triggering?

* **What is actually rewritten?** Can you provide layer-wise and head-wise analyses of rewrite magnitudes (gate values, ΔK/ΔV norms), and their correlation with downstream accuracy, to demonstrate nontrivial, stable edits?

* **Sensitivity to selection policy**: Beyond top-k by attention mass, evaluate (a) per-layer k, (b) diversity-regularized selection, (c) recency vs. salience trade-offs, and (d) dynamic k conditioned on step difficulty. Table 2 hints that MATH prefers larger k. Can adaptive k close the MATH gap?

---

> ### Author Response · Authors · 2025-11-24
> **Response to reviewer cuPq (part 1)**
>
> We thank the reviewer for the thoughtful and constructive comments. Below we address the listed weaknesses and questions in turn, and indicate where we have added clarifications or new analyses in the revised version.
>
> For **Weakness 1 (scope and novelty w.r.t. cache operators)**, we have clarified in the related-work section how our method differs from existing cache-operator families. Compression-oriented cache operators are designed primarily to reduce memory footprint for long contexts, aiming to preserve performance while discarding information; they do not explicitly target improved reasoning quality and typically operate by evicting, merging, or down-projecting cache entries. Other ALSC-style methods operate in the same paradigm as vanilla LLMs by appending extra token-like vectors to the cache (latent rollouts, pause tokens, differentiable cache augmentation), effectively lengthening the sequence. In contrast, our Cache Processor performs in-place rewrites of selected KV entries at fixed dimensionality with the explicit goal of improving sequence-level reasoning, closer to consolidation/reconsolidation than to compression. We state this distinction more explicitly and emphasise that our contribution is to introduce cache reconsolidation as a distinct architectural primitive that can be composed with token-mediated schemes, rather than a minor variant of existing operators.
>
> Regarding **Weakness 2 (supervision signal and IB framing)**, we agree that training the Processor only via next-token cross-entropy with truncated BPTT, without an explicit IB or MI-based auxiliary objective, is a significant limitation of the current architecture.We position the IB analysis explicitly as a conceptual lens rather than as an optimised variational IB objective, and we note that direct MI estimation over full KV caches in long sequences is technically challenging and computationally expensive in our setting. Our primary goal in this work is to demonstrate that structured cache rewrites can already yield benefits under this suboptimal supervision regime; we use held-out performance together with direct measurements of nontrivial KV rewrites (see below) as indirect evidence for improved predictive efficiency. Designing future variants with stronger supervision signals or tractable IB-style objectives is an important direction that we plan as follow-up work.
>
> For **Weakness 3 (mixed results on OOD / symbol-heavy tasks)**, we would like to clarify that Bottlenecked Transformers tend to underperform SFT only on LogiQA and Gaokao, and even there not consistently across backbones. We agree that this pattern suggests our current Processor architecture is relatively simple and somewhat sensitive to hyperparameters and domain specifics at this stage. In the paper we will clarify this limitation and explicitly point out that improving robustness on OOD tasks via better consolidation schedules and more adaptive hyperparameters is a core goal for future variants.
>
> Addressing **Question 1 (ablations on triggers)**, we agree that newline is not a universal proxy for reasoning steps. In our benchmarks, however, the supervision data are formatted so that each line corresponds to a distinct intermediate step in the chain-of-thought, so newline boundaries closely align with step boundaries; we now make this design choice explicit and present the “newline trigger” as a dataset-specific heuristic. Beyond this, we include a new on study invocation frequency via an ablation over the recent-step window $R$, where we invoke the Processor once at the end of the prompt and then every $R$ tokens; performance remains robust over a broad range of $R$, with mild gains for moderate windows. We agree that prediction-error/surprise-based triggers, punctuation-based schemes, or learnable triggering are natural extensions, but we view them as orthogonal to the core contribution of introducing cache reconsolidation and leave them to follow-up work.

---

> > ### Author Response · Authors · 2025-11-24
> > **Response to reviewer cuPq (part 2)**
> >
> > In response to **Question 2 (what is actually rewritten?)**, we thank the reviewer for pointing out our lack of analysis on update magnitudes. We have added analyses that quantify rewrite magnitudes across layers and heads, which has uncovered interesting findings. On GSM8K with the Llama-3.2-1B Bottlenecked Transformer, we measure cosine distances between KV entries before and after each Processor invocation, separately for the recent-step window $R$, the top-$k$ recalled tokens, and all rewritten entries. We find that value vectors undergo consistent, nontrivial updates, while key vectors change very little, indicating that the Processor primarily reshapes content rather than the addressing scheme. Layer–head heatmaps show that edits are concentrated in earlier layers and stabilise after a small number of invocations, suggesting a stable, non-degenerate rewrite regime.
> >
> > Finally, for **Question 3 (sensitivity to selection policy and adaptive $k$)**, we have not yet run additional experiments beyond our fixed-$k$ settings. Empirically, we observe that MATH tends to prefer larger $k$, which we hypothesise is due to its more complex problems and longer reasoning traces requiring a broader set of recalled tokens. We agree that adaptive $k (for example, varying with step difficulty or layer) is a very interesting direction that we had not considered. While we have not explored it in this work, since our focus was on validating a basic consolidation mechanism, we plan to investigate adaptive selection policies in future variants and we thank the reviewer for this suggestion.
> >
> > We hope these clarifications, additional analyses, and explicit statements of scope address the reviewer’s concerns, and we are grateful for the suggestions that help sharpen both the positioning and interpretation of our results.

---

> > > ### Comment · Reviewer_cuPq · 2025-11-25
> > >
> > > Thank you for your responses. I decide to maintain my positive score.

---

### Official Review · Reviewer_8HNH · 2025-10-31

**Soundness:** 3
**Presentation:** 4
**Contribution:** 3
**Rating:** 4
**Confidence:** 4

**Summary:**

This paper proposes Bottlenecked Transformers, which add a lightweight non-causal Cache Processor that periodically rewrites the KV cache at newline-delimited “reasoning step” boundaries to consolidate recent tokens and reconsolidate top‑k recalled past tokens for better sequence-level reasoning. The approach is motivated via an Information Bottleneck perspective arguing vanilla decoder-only Transformers over-retain history in the KV cache and thus benefit from in‑place memory edits that preserve predictive information without compressing dimensionality. Concretely, the processor performs periodic, in-place global rewrites of recent KV segments and selected prior entries conditioned on the latest context, acting as sequence-level auxiliary latent-space computation adjacent to standard decoding. Across seven mathematical reasoning benchmarks and four backbone LLMs, the method yields consistent gains over vanilla and pause-token baselines, with improvements up to +6.6 percentage points on selected tasks and models.

**Strengths:**

1. Clear problem framing of ALSC at the sequence level and principled positioning of cache rewriting as consolidation/reconsolidation rather than compression, with an architecture that is simple to integrate and keeps KV dimensionality unchanged.​

2. Information-theoretic perspective highlights a plausible failure mode of vanilla decoder-only training (KV states retain unnecessary sequence detail), and motivates a targeted non-causal rewrite to increase predictive efficiency without shrinking memory footprint.​

3. Consistent improvements over SFT and “SFT + pause tokens” across multiple backbones and benchmarks, including sizeable gains on in-distribution math tasks (e.g., +6.6 on SVAMP with Llama‑3.2‑1B; +4.6 on GSM8K with Llama‑3.2‑3B) and thoughtful ablations on epoch-matched budgets and the reconsolidation window k.​

4. Implementation details are thorough (selection policy via attention mass, layer-aligned processor blocks, gated residual rewrites), with reproducibility notes and multiple architectural/training ablations provided.​

**Weaknesses:**

1. Theorem 4.2 provides a lower bound linking token cross‑entropy to a sum of mutual information terms, but the text then treats autoregressive training as “maximizing both” $$I(S_{0:n};\hat Z)$$ and $$I(\hat Z;S_{n+1})$$, which does not follow from a loose bound and is not shown to hold per‑term, weakening the justification for the proposed remedy.
2. The “terminal bottleneck” claim for the KV cache plus last hidden state is used to argue that the cache retains reconstructive detail that impedes generalization, yet no empirical evidence is provided that the cache can reconstruct inputs or that rewrites reduce $$I(X;Z)$$ while preserving $$I(Z;Y)$$ as required by the IB narrative.
3. No IB‑style loss or MI estimation is used; reliance on SGD noise and the data‑processing inequality is asserted as a path to compression, but there is no quantification of predictive efficiency $$I(Z;Y)/I(X;Z)$$ before vs. after cache rewrites to support the core thesis.
4.  Comparisons omit strong ALSC and cache‑operator baselines beyond pause tokens, such as latent rollouts, differentiable cache augmentation, or modern cache merging/compression tuned for reasoning, leaving performance advantages over alternative designs unclear under matched budgets.
5. The authors notes reconsolidation may require prediction‑error gating, but does not test such triggers or analyze boundary sensitivity (e.g., recent‑step window length R) in the main results.
6.  Results appear single‑seed with no confidence intervals or significance testing, and there is no contamination analysis despite training on synthetic math mixes that may overlap benchmark styles
7. Training the processor incurs ~6× memory and ~20× wall‑clock over SFT, while inference adds ~25% memory and ~45% latency despite infrequent invocation, which is a significant overhead for modest accuracy gains on small/medium backbones.
8. There are no MI proxies/estimates, ablations that track retrieval fidelity vs. abstraction, or causal analyses showing which information is discarded/retained by the processor.
9. No layer/head‑level probes, attention‑pattern shifts, or KV‑state visualizations that explain why and when reconsolidation helps or hurts (e.g., MATH vs. Gaokao outcomes).

**Questions:**

1. The IB narrative relies on reducing $I(X;Z)$ while keeping or improving $I(Z;Y)$; can the authors either add a tractable variational IB objective (e.g., CPC/InfoNCE-style bounds) or present calibrated MI proxies that show the intended movement along the efficiency frontier, rather than only accuracy deltas ?
2. I suggest the authors include strong latent and cache-operator baselines under matched compute, add non‑math and multilingual tasks, and report multi‑seed CIs and contamination checks to substantiate generalization claims
3. Can the authors run sensitivity analyses for $k$, $R$, and invocation frequency, and report compute-normalized outcomes to justify overheads.
4. I am curious, if training from scratch with a joint backbone‑processor objective to test whether the “frozen backbone” constraint is the main limiter, and whether end‑to‑end training reaches better predictive efficiency under equal compute?

---

> ### Author Response · Authors · 2025-11-24
> **Response to reviewer 8HNH (part 1)**
>
> We sincerely thank the reviewer for the careful and constructive feedback, which has helped us substantially improve the clarity, positioning, and empirical support of the paper. Below we group our responses by theme and explicitly indicate which Weaknesses and Questions each section addresses.
>
> ## A. Information-bottleneck framing and Theorem 4.2
> *(addresses Weaknesses 1–3, 8 and Question 1)*
>
> ### A1. Theorem 4.2 and “maximisation” wording (Weakness 1)
> We agree that describing autoregressive training as “maximising” the mutual information terms was imprecise given the looseness of the bound. In the revision we now make a slightly looser claim that training via next-token loss should **encourage high mutual information**, rather than strictly maximises it. Our derivation does, however, provide bounds for the two MI terms separately. As detailed in the proof (Eqs. 19 and 26), the autoregressive loss \(L(\theta)\) satisfies:
>
> 1. **History Retention:**
>
> $L(\theta) \leq \sum_{n} [I(S_{0:n}; C_{0:n}) - H(S_{n+1}\mid S_{0:n})]$
>
> 2. **Prediction:**
>
> $L(\theta) \leq \sum_{n} [I(C_{0:n}; S_{n+1}) - H(S_{n+1})]$
>
> Since the entropy terms depend only on the data distribution, minimising $L(\theta)$ is consistent with *encouraging* both high $I(S_{0:n}; C_{0:n})$ and high $I(C_{0:n}; S_{n+1})$. We thank the reviewer for their helpful suggestion in pointing out this lack of clarity.
>
> ### A2. “Terminal bottleneck” and reconstruction intuition (Weakness 2)
>
> We thank the reviewer for pointing out lack of clarity on this point, and agree that our original wording around “reconstructing the input” could be read as claiming the existence of an explicit inverse map from KV states to tokens, which is stronger than intended. We have softened this language in the paper and clarify the operational intuition here.
>
> Let $t$ index tokens, with $S_{0:t}$ a token prefix and $C_{0:t}$ the KV cache plus final hidden state after processing that prefix. Under teacher-forced training, for each $i < t$ the model computes $C_{0:i}$, sufficient (given the parameters) to form $p_\theta(S_{i+1} \mid C_{0:i})$. When we process the full prefix $S_{0:t}$, the final cache $C_{0:t}$ stores entries for positions $0,\dots,t$, and the sub-cache up to position $i$ coincides with the earlier $C_{0:i}$. Thus $C_{0:t}$ contains a *predictive trace* of the right-shifted sequence $(S_1,\dots,S_t)$ in the sense that it preserves information about next-token predictors $\{p_\theta(S_{i+1} \mid C_{0:i})\}_{i<t}$, rather than a compressed summary. This is the sense in which we referred to “reconstructive detail.”
>
> ### A3. IB narrative, MI estimation, and indirect evidence (Weaknesses 3, 8, 9 and Question 1)
>
> Estimating mutual information over the full KV cache is challenging: the representation is high-dimensional, continuous, and variable-length, and robust variational bounds (e.g., CPC/InfoNCE-style) would require additional encoders, careful negative sampling, and substantial extra compute. For this reason, we chose not to introduce a potentially unstable MI estimator and instead use IB as a conceptual lens rather than an explicit training objective. Instead, we provide indirect empirical evidence consistent with movement towards a more efficient IB trade-off:
>
> 1. **Structured cache rewrites.**
>
> In new experiments (Sec. 6.5, “Processor Rewrite Magnitudes”), we measure cosine distances between KV entries before and after Processor calls on GSM8K with Llama-3.2-1B Bottlenecked Transformers. We report mean distances separately for: (i) top-$k$ recalled tokens, (ii) the recent-step window, and (iii) all rewritten entries. We find:
>
>     * Value vectors undergo systematic, nontrivial updates,
>     * Key vectors remain almost unchanged,
>     * Edits are largest at early invocations and then stabilise, and
>     * Changes are concentrated in early layers, with little change in middle/later layers.
>
> This indicates the Processor is not collapsing to the identity but rewrites the *contents* of working memory while mostly preserving the addressing scheme.
>
> 2. **Predictive information via test performance.**
>
> From $I(Y;Z) = H(Y) - H(Y \mid Z)$ and fixed $H(Y)$, improved held-out performance (lower NLL / higher accuracy) implies more predictive information in $Z$ about $Y$. Across seven reasoning benchmarks and four backbones, Bottlenecked Transformers consistently outperform SFT and SFT+pause-token baselines under matched training budgets, suggesting that $C'_{0:n}$ is more informative about future tokens than the unprocessed cache.
>
> We see robust MI estimation for full KV caches as valuable future work but beyond the scope of this paper, and we now state more explicitly that our empirical evidence is qualitative support for the IB narrative rather than a direct MI measurement.

---

> > ### Author Response · Authors · 2025-11-24
> > **Response to reviewer 8HNH (part 2)**
> >
> > ## B. Baselines, scope, generalization, and contamination
> >
> > *(addresses Weaknesses 4, 6 and Question 2)*
> >
> > ### B1. ALSC and cache-operator baselines (Weakness 4)
> >
> > We agree it is important to position Bottlenecked Transformers relative to other ALSC [families.In](http://families.in/) the original submission we compared against SFT and SFT + pause tokens as a widely used token-mediated baseline. Following the reviewer’s suggestion, we have added a **latent rollout** baseline that inserts a short latent micro-sequence between reasoning-line tokens, trained under the same data mix and comparable compute. On our math benchmarks, latent rollouts underperform both pause tokens and Bottlenecked Transformers, consistent with similar reports (e.g., Coconut) in small/medium-backbone regimes.
> >
> > Conceptually, pause tokens, latent rollouts, and differentiable cache augmentation are all **token-mediated**: they append extra token-like vectors to the cache and lengthen the attended sequence. Our Cache Processor instead performs **in-place** rewrites of existing KV entries at fixed cache dimensionality. These approaches are orthogonal, and Bottlenecked Transformers could be composed with token-mediated schemes by allowing the Processor to rewrite both original and appended entries.
> >
> > We did not include compression-oriented cache operators and recurrent memory architectures as baselines because they are primarily tuned for long-context compression and memory footprint reduction, whereas our setting involves relatively short math contexts where memory is not the limiting factor. Applying such methods fairly would require substantial re-tuning for reasoning generalisation rather than compression, which we view as out of scope and a new method in and of itself. We note that combining Bottlenecked Transformers with long-context cache-compression schemes is interesting future work.
> >
> > ### B2. Single-seed runs, confidence, and contamination (Weakness 6 and Question 2)
> >
> > Because Bottlenecked Transformer training is expensive (see response to Weakness 7), all reported runs are single-seed. To partially compensate, we evaluate across seven benchmarks and four backbones and consistently see improvements over SFT and SFT+pause tokens, including sizeable margins on GSM8K and SVAMP. This makes it less likely that our main conclusions are artefacts of a particular seed, but we acknowledge the limitation and now state it explicitly.
> >
> > Regarding contamination: the synthetic math mixtures are constructed only from the *training* splits of GSM8K, MATH, and related datasets; official test splits are never used for training. While stylistic overlap is possible given the domain, there is no direct inclusion of test examples. We have clarified this and acknowledge that we do not perform a full contamination audit; broader, non-math tasks and multi-seed evaluations are left for future work.

---

> ### Author Response · Authors · 2025-11-24
> **Response to reviewer 8HNH (part 3)**
>
> ## C. Design choices, sensitivity, compute, and backbone training
>
> *(addresses Weaknesses 5, 7 and Questions 3–4)*
>
> ### C1. Boundary sensitivity, $k$, $R$, and invocation frequency (Weakness 5 and Question 3)
>
> We agree prediction-error gating is a natural extension, but our goal in this work was to first test whether periodic consolidation is beneficial at all using the simplest unconditional Processor. Whilst prediction error gating is in our opinion a very interesting avenue that we aim to explore in future work, it is not central to the core idea of consolidation as we define it in this paper.
>
> We have added a recent-step window $R$/invocation-frequency ablation (Sec. 6.4), where we invoke the Processor once at the end of the prompt and then every $R$ tokens. Across tasks, performance is broadly stable over a wide range of $R$, with mild gains for moderate windows ($R \in [64,96]$) and only small drops when consolidation is restricted to very short windows. Together with our existing top-$k$ reconsolidation ablation, this suggests robustness to the exact boundary schedule, provided consolidation operates over a medium-horizon segment of the cache.
>
> ### C2. Training and inference overheads (Weakness 7)
>
> We agree that the current training and inference overheads are substantial and represent a limitation of our reference design. Much of this overhead is due to implementation constraints: global cache rewrites require chunk-wise teacher forcing, and our Processor cannot currently use FlashAttention, whereas SFT baselines do. Our ablations over $R$ and top-$k$ indicate that effective consolidation does *not* require frequent invocations or maximal windows, suggesting room to reduce overhead with more optimised architectures and kernels that integrate consolidation with efficient attention. Whilst our aim for this paper was to explore whether simple cache consolidation was beneficial, a large aim for our future work is to iron out these computational efficiencies.
>
> ### C3. End-to-end vs frozen-backbone training (Question 4)
>
> In response to **Question 4**, we did experiment with end-to-end training where the backbone is unfrozen. In preliminary trials this destabilised the backbone and led to worse performance than our frozen-backbone setup. We have added this observation to the paper. Stabilising joint backbone–processor training and exploring whether it can yield further gains under equal compute is an interesting but distinct challenge that we leave for future work.
>
> We again thank the reviewer for the detailed and constructive comments, which have directly guided multiple clarifications and new analyses. We believe these changes strengthen the paper and better situate Bottlenecked Transformers within the broader ALSC landscape.

---

### Author Response · Authors · 2025-11-24
**Response to all reviewers**

We would like to extend a warm thanks to all reviewers for their thoughtful, detailed, and constructive feedback. The comments were somewhat consistent across reviews and helped us clarify both the theoretical claims and the empirical positioning of Bottlenecked Transformers. In particular, concerns around the soundness and interpretation of our IB theorem, the strength of baselines, the consolidation schedule, and the nature of the cache rewrites directly motivated the following core changes:

- **Clarified and refined the main mutual-information theorem:**

    In response to questions about the soundness and interpretation of our IB framing, we replaced the earlier, looser joint bound with the new Theorem 4.2, which provides two separate bounds on the autoregressive objective $L(\theta)$ in terms of $\sum_n I(S_{0:n}; C_{0:n})$ and $\sum_n I(C_{0:n}; S_{n+1})$ plus fixed entropy terms. We now explicitly state that, under fixed data entropies, maximising $L(\theta)$ *encourages* increases in both mutual-information components, and we add a concise tokenwise explanation showing that $C_{0:t}$ contains all earlier caches $\{C_{0:i}\}_{i<t}$ and thus encodes a high-fidelity predictive trace of the right-shifted tokens rather than a single compressed summary.

- **Added a latent rollout baseline:**

    One reviewer asked for stronger ALSC/cache-operator comparisons beyond pause tokens. To address this, we added a latent rollout baseline under a matched compute budget and now compare Bottlenecked Transformers against SFT, SFT+pause tokens, and latent rollouts. This positions our method more clearly within the existing ALSC design space and helps separate the benefits of in-place cache consolidation from those of more conventional token-mediated latent computation.

- **Ablated the recent-step window \(R\) (RSW) and invocation frequency:**

    Concerns about the simplicity of our newline trigger and questions about boundary sensitivity motivated a new ablation over the recent-step window $R$, where we invoke the Processor once at the end of the prompt and then every $R$ tokens. We show that performance is broadly robust over a reasonable range of $R$, with moderate windows working best, which helps demonstrate that our gains are not tied to a single brittle schedule and provides a clearer picture of the compute/benefit trade-off.

- **Analysed KV update magnitudes and rewrite structure:**

    Reviewers also asked “what is actually being rewritten?” and requested evidence that our method goes beyond a near-identity edit of the KV cache while remaining stable. In response, we added a rewrite-magnitude analysis across layers and heads. These results show that the Processor performs structured, nontrivial, and primarily value-focused edits that concentrate in earlier layers and stabilise over time, directly supporting our reconsolidation interpretation and complementing the IB-motivated discussion.


These changes can be found in the updated PDF. Please see responses to individual reviews below. We are very grateful for the reviewers’ insights, which we believe have allowed us to  substantially improved the clarity, positioning, and empirical support of the paper.

---

### Meta-Review · Area_Chair_YofD · 2026-01-14

**Summary:**

This paper argues that periodically reprocessing the model’s working memory (KV cache) should aid generalization from the lens of Information Bottleneck theory . The paper introduces Bottlenecked Transformers,  a method that periodically consolidates the KV cache using a cache processor. The reviews for the paper were mixed with one reviewers giving 4, 6 and 8.

There were several concerns raised by the reviewers. This included large training and inference overhead of the approach for very modest gains, somewhat limited novelty and heuristic trigger. The paper has interesting ideas but my biggest concern about this paper is that of significant overhead for very modest gains. This concern was uniformly raised by the reviewers but the author response is not satisfactory in my opinion. Furthermore, it is unclear if the gains hold at scale (e.g. Llama 3.1 8B in Table 1 already look somewhat weak). I think this paper would greatly benefit from addressing these concerns before publication.

**Reviewer Concerns:**

Reviewer 8HNH raised a valid concern that the method incurs a massive cost—training is approximately 20x slower and inference is ~45% slower (on a 1B model) compared to standard SFT. This makes the method currently impractical for large-scale deployment. In my opinion, the performance improvement, especially at larger scale is still not large to justify this addition cost. This point was also acknowledged by the authors. I think this is probably one of the major weakness of the paper. The reviewer also added requested for missing baseline, which was added during the rebuttal.

Reviewer cuPq raised concern about the method being similar to the existing cache compression operators. The authors clarified the distinction between consolidation (improving reasoning quality) and compression (reducing memory footprint).

**Reviewer Scores:**

Reviewer 8HNH would probably keep their original score. The overhead concern is actually very valid and the authors did not address this concern sufficiently.

Reviewer cuPq & kt4M would probably maintain their positive score. Overall, I think the paper is on the borderline. I think the question of overhead is important to address in my opinion. Reviewer kt4M score of 8 is probably on the higher end given this important concern.

---

### Decision · Program_Chairs · 2026-01-26

Accept (Poster)